# Peer review of "Determination of VOCs in Surgical Resected Tissues from Colorectal Cancer Patients by Solid Phase Microextraction Coupled to Gas Chromatography–Mass Spectrometry"

_applsci, doi:10.3390/app11156910_

Round 1

Reviewer 1 Report

This method is very simple and important for detecting colorectal cancer.

However, I have few questions

  1. In this MS, only 3 patients are analyzed, I suggest to analyze at least 6 patients to confirm whether the method is stable.
  2. These VOCs including benzaldehyde, benzoic acid, dodecane, ethylbenzene, octanol, tetradecane and toluene were selected as target compounds. Does it have the same composition in other cancer tissues, such as lung adenocarcinoma tissue.

Author Response

This method is very simple and important for detecting colorectal cancer.

However, I have few questions:

Q1: In this MS, only 3 patients are analyzed, I suggest to analyze at least 6 patients to confirm whether the method is stable.

A.1: In principle, the greater the number of patients analyzed, the more reliable the results would be. However, the current number of subjects analyzed still clarifies the potential of the method.

Q.2: These VOCs including benzaldehyde, benzoic acid, dodecane, ethylbenzene, octanol, tetradecane and toluene were selected as target compounds. Does it have the same composition in other cancer tissues, such as lung adenocarcinoma tissue.

A.2: The target compounds have been selected basing on results of our previous work on colorectal cancer. As far as, we know these VOCs, although they belong to compound classes common to other cancer tissues [Altomare DF, Di Lena M, Porcelli F, Trizio L, Travaglio E, Tutino M et al. Exhaled volatile organic compounds identify patients with colorectal cancer. Br J Surg (2013) 100: 144–150; de Meij TG, Larbi IB, van der Schee MP, Lentferink YE, Paff T, Terhaar sive Droste JS et al. Electronic nose can discriminate colorectal carcinoma and advanced adenomas by fecal volatile biomarker analysis: proof of principle study. Int J Cancer (2014) 134: 1132–1138; Wang C, Ke C, Wang X, Chi C, Guo L, Luo S et al. Noninvasive detection of colorectal cancer by analysis of exhaled breath. Anal Bioanal Chem (2014) 406: 4757–4763; Amal H, Leja M, Funka K, Lasina I, Skapars R, Sivins A et al. Breath testing as potential colorectal cancer screening tool. Int J Cancer (2016) 138: 229–236; Bhattacharyya D, Kumar P, Mohanty SK, Smith YR, Misra M. Detection of four distinct volatile indicators of colorectal cancer using functionalized titania nanotubular arrays. Sensors (2017) 17: 1795; Kort S, Tiggeloven MM, Brusse-Keizer M, Gerritsen JW, Schouwink JH, Citgez E et al. Multi-centre prospective study on diagnosing subtypes of lung cancer by exhaled-breath analysis. Lung Cancer (2018) 125: 223–229; Marzorati D, Mainardi L, Sedda G, Gasparri R, Spaggiari L, Cerveri P. A review of exhaled breath: a key role in lung cancer diagnosis. J Breath Res (2019) 13: 034001; Kumar S, Huang J, Abbassi-Ghadi N, Mackenzie HA, Veselkov KA, Hoare JM et al. Mass spectrometric analysis of exhaled breath for the identification of volatile organic compound biomarkers in esophageal and gastric adenocarcinoma. Ann Surg (2015) 262: 981–990; Markar SR, Wiggins T, Antonowicz S, Chin ST, Romano A, Nikolic K et al. Assessment of a noninvasive exhaled breath test for the diagnosis of oesophagogastric cancer. JAMA Oncol (2018) 4: 970–976; Markar SR, Brodie B, Chin ST, Romano A, Spalding D, Hanna GB. Profile of exhaled-breath volatile organic compounds to diagnose pancreatic cancer. Br J Surg (2018) 105: 1493–1500; Phillips M, Cataneo RN, Ditkoff BA, Fisher P, Greenberg J, Gunawardena R et al. Prediction of breast cancer using volatile biomarkers in the breath. Breast Cancer Res Treat (2006) 99: 19–21], revealed high sensitivity and specificity for colorectal cancer patients [D. F. Altomare, A. Picciariello, M. T. Rotelli, M. De Fazio, A. Aresta, C. G. Zambonin, L. Vincenti, P. Trerotoli, N. De Vietro, Chemical signature of colorectal cancer: case–control study for profiling the breath print, BJS Open 4 (2019), 1189-1199], confirming, in conclusion, that each cancer type is associated with a unique molecule pattern.

Reviewer 2 Report

In this study De Vietro and colleagues demonstrated a new approach of utilizing solid phase microextraction coupled to gas chromatography-mass spectrometry in determining the volatile organic compounds (VOCs) in surgical resected tissues from colon rectal cancer patients. The experimental data presented are solid and the authors’ data analysis and interpretation are convincing. This approach would be important and useful for improving early diagnosis of colorectal cancer.

To improve the manuscript quality, I would like to suggest the authors to do revision work in the following aspects:

1) One component in the abstract is missing. That is the implication or significance of their work.

2) The authors introduced an abbreviation, “VOCs”, in the abstract without the full term, “volatile organic compounds”.

3) Image resolutions for the Figures 2, 5 and 6 should be enhanced.

4) Various errors of typing, punctuation and grammar exist in the text, e.g., i) “the whole procedures fully validated” in the abstract; ii) “together with other important variables such as, ionic strength, pH, sample volume, uniformity and stirring speed [22-24]” in the introduction; iii) “(one men and two women, mean age of 65 ± 10 years, cancer located in right colon, stage III)” in the Materials and methods; iv) “(30 m x0.25 mm i.d.,0.25 mm film thickness)” in the Materials and methods; v) “RSD % values were always ≤ 10,5 % for all analytes at all tested concentrations, while RSD % values ≤ 9, 9 % were obtained working with the DI approach” in the Results and discussion.

Author Response

In this study De Vietro and colleagues demonstrated a new approach of utilizing solid phase microextraction coupled to gas chromatography-mass spectrometry in determining the volatile organic compounds (VOCs) in surgical resected tissues from colon rectal cancer patients. The experimental data presented are solid and the authors’ data analysis and interpretation are convincing. This approach would be important and useful for improving early diagnosis of colorectal cancer.

To improve the manuscript quality, I would like to suggest the authors to do revision work in the following aspects:

Q.1: One component in the abstract is missing. That is the implication or significance of their work.

A.1: The missing component (octanal instead of octanol) was added in the abstract.

Q.2: The authors introduced an abbreviation, “VOCs”, in the abstract without the full term, “volatile organic compounds”.

A.2: Correction done.

Q.3: Image resolutionsfor the Figures 2, 5 and 6 should be enhanced.

A.3: Correction done. The character dimension has been increased and the image resolution has been passed from 300 dpi to 600 dpi (see files attatched).

Q.4: Various errors of typing, punctuation and grammar exist in the text, e.g., i) “the whole procedures fully validated” in the abstract; ii) “together with other important variables such as, ionic strength, pH, sample volume, uniformity and stirring speed [22-24]” in the introduction; iii) “(one men and two women, mean age of 65 ± 10 years, cancer located in right colon, stage III)” in the Materials and methods; iv) “(30 m x0.25 mm i.d.,0.25 mm film thickness)” in the Materials and methods; v) “RSD % values were always ≤ 10,5 % for all analytes at all tested concentrations, while RSD % values ≤ 9, 9 % were obtained working with the DI approach” in the Results and discussion.

A.4: Corrections done, replacing the previous sentences with the following:

i) The procedures fully validated; ii) together with other variables, including ionic strength, pH, sample volume, uniformity and stirring speed; iii) one man and two women, mean age 65 ± 10 years, tumor localized to the right colon, stage III; iv) 30 m x0.25 mm i.d.,0.25 mm film thickness; v) RSD % values ≤ 10,5   (headspace) and ≤ 9, 9 % (direct immersion) were always obtained for all the analytes at different concentrations levels.